# Learning Optimal Reserve Price against Non-myopic Bidders

**Zhiyi Huang**[*]          **Jinyan Liu**          **Xiangning Wang**

Department of Computer Science
The University of Hong Kong
{zhiyi, jyliu, xnwang}@cs.hku.hk

## Abstract

We consider the problem of learning optimal reserve price in repeated auctions against non-myopic bidders, who may bid strategically in order to gain in future rounds even if the single-round auctions are truthful. Previous algorithms, e.g., empirical pricing, do not provide non-trivial regret rounds in this setting in general. We introduce algorithms that obtain a small regret against non-myopic bidders either when the market is large, i.e., no single bidder appears in more than a small constant fraction of the rounds, or when the bidders are impatient, i.e., they discount future utility by some factor mildly bounded away from one. Our approach carefully controls what information is revealed to each bidder, and builds on techniques from differentially private online learning as well as the recent line of works on jointly differentially private algorithms.

## 1 Introduction

The problem of designing revenue-optimal auctions based on data has drawn much attention in the algorithmic game theory community lately. Various models have been studied, notably, the sample complexity model [10, 13, 23, 32, 31, 36, 12, 18, 8] and the online learning model [7]. However, these existing works all implicitly assume that bidders are myopic in the sense that they will faithfully report their valuations as long as the mechanism used in each round is truthful, without considering how their bids may affect 1) the choices of mechanisms and 2) the behaviors of other bidders in future rounds in which they may also participate. What happens in the presence of non-myopic bidders?

> **Example.** Suppose a seller has a fresh copy of the good for sale every day, where its value for any bidder is bounded between $0$ and $1$. The seller sets a price at the beginning of each day. Then, a bidder (say, randomly chosen from a large yet finite pool of potential bidders) arrives and submits a bid: If the bid is higher than the price of the day, he gets the item and pays the price. Further, suppose the seller adopts the solution proposed by the sample complexity literature and decides to set the price at $0.5$ on day $1$, and in each of the following days to use the empirical price, i.e., the best fixed price w.r.t. the bids in previous days.

If bidders are myopic, bidding truthfully is a dominant strategy since the mechanism on each day is effectively posting a take-it-or-leave-it price. As a result, the seller will be able to converge to the optimal fixed price w.r.t. to the pool of potential bidders.

If bidders are non-myopic, however, their strategies are more intriguing. A bidder may underbid whenever the empirical price of the day (which is deterministic) is higher than his value, inducing the same result of not winning in the current round but leads to lower future prices compared with

---

[*]Supported in part by an RGC grant HKU17257516E.

truthful bid. By the same reasoning, even if the bidder wants to win the current copy of the good, he will not bid truthfully; instead, he will submit a bid that equals the current price. Due to the strategic plays of non-myopic bidders, the seller may in fact converge to a price close to $0$. The take-away message of this example is that directly applying existing algorithms to scenarios where bidders are non-myopic could be a disaster in terms of revenue. Hence, we ask:

*Are there learning algorithms that work well even in the presence of non-myopic agents?*

In particular, can we extend the online learning model to allow non-myopic bidders, and design algorithms that provably have small regret against the best fixed auction?

## 1.1 Our Results and Techniques

Our main contribution is a positive answer to the above question, subject to one of the following two assumptions: Either the bidders are impatient in the sense that they discount utility in future rounds by some discount factor (mildly) bounded away from $1$ (*impatient bidders*), or any bidder comes in only a small fraction of the rounds (*large market*). The assumptions are relatively natural, and are further necessary: If the same bidder appears everyday without discounting future utility, no algorithms can guarantee non-trivial regret (e.g., Theorem 3 of Amin et al. [3]).

**Single-bidder.** Let us first consider the case in the example, where only a single bidder comes on each day. We show the following:

**Informal Theorem 1.** *For any $\alpha \in (0, 1)$, our online pricing algorithm has regret at most $\alpha T$ against non-myopic bidders when $T \geq \tilde{\mathcal{O}}(\alpha^{-4})$, and either impatient bidders or large market holds.*[2]

This is equivalent to a sub-linear $\tilde{\mathcal{O}}(T^{3/4})$ regret bound. Here, we omit in the big-O notation a term that depends on either the discount factor or the maximum number of rounds that a bidder can appear. (It holds when the number of rounds that a bidder can participate in is at most $\alpha^4 T$, or when the discount factor is at most $1 - \frac{1}{\alpha^4 T}$. See Theorem 3.1 for the formal statement.) In addition, we omit log factors in the $\tilde{\mathcal{O}}$ notation.

Typical mechanism design approach may seek to design online learning mechanisms such that truthful biddings form an equilibrium (or even better, are dominating strategies) even if bidders are non-myopic, and to get small regret given truthful bids. However, designing truthful mechanisms that learn in repeated auctions seems beyond the scope of existing techniques.

We take an alternative path by relaxing the incentive property: We aim to ensure that bidders would only submit "reasonable" bids within a small neighborhood of the true values. Note that the notion of regret is robust to small deviations of bids. Applying an online learning algorithm on "reasonable" bids (instead of truthful bids) results in only a small increase in the regret.

To explain how to achieve the relaxed incentive property, first consider why a bidder would lie. Since the single-round auctions are truthful, a bidder can never gain in the current round by lying. Lying is preferable only if the future gain outweighs the current loss (if any). The seller's algorithm in the example, i.e., using empirical prices, suffers on both ends. On one hand, lying has no cost when bid and true value are both greater (or both smaller) than the price. On the other hand, the current bid has huge influence on future prices; in particular, the first bid dictates the price in the second round.

We will design online auctions such that (1) deviating too far from the true value in the current round is always costly, and (2) the influence of the current bid on future prices/utilities is bounded. Achieving the first property turns out to be easy. Note that lying has a cost whenever the price falls between the bid and true value. On each day, with some small probability our mechanisms pick the price randomly to ensure that the price has a decent probability to fall between the value and any "unreasonable" bid that deviates a lot.

The second property may seem trivial through the following incorrect argument: Online learning algorithms, e.g., multiplicative weight, are intrinsically insensitive to the bid on any single day and, thus, satisfy the second property. The argument incorrectly assumes subsequent bids will remain the same regardless of the current bid, omitting that they are controlled by strategic bidders and, thus, are affected by the current bid through its influence on subsequent prices. We use an implementation of the follow-the-perturbed-leader algorithm based on the tree-aggregation technique [2] from differential

privacy. Note that due to the above reasoning, differential privacy does not imply the second property. Nevertheless, we show that the algorithm in fact satisfies a slightly stronger guarantee than differential privacy, which is sufficient for our proof.

Our techniques further allow us to get essentially the same bound even in the bandit setting, i.e., seller observes if the bidder buys the copy but not his bid. This is shown in full version.

**Multi-bidder.** Our approach can be extended to obtain positive results with $n > 1$ bidders and $m > 1$ copies of the good per round, with some extra ingredients we shall explain shortly.

**Informal Theorem 2.** *For any $\alpha \in (0, 1)$, our algorithm runs an approximate version of Vickrey with an anonymous reserve price on each day with regret at most $\alpha m T$ against the best fixed reserve price if (1) $T \geq \tilde{O}(\frac{n}{m\alpha^{4.5}})$, (2) $m \geq \tilde{O}(\frac{\sqrt{n}}{\alpha^3})$, (3) either impatient bidders or large market holds.*

Simply running a follow-the-perturbed-leader algorithm with tree-aggregation as in the single-bidder case does not work in the multi-bidder setting because a bidder's current bid can now affect other bidders' subsequent bid through the allocations and payments in the current round. We need $m > 1$ in the multi-bidder setting due to the use of joint differential privacy to control the influence among bidders. It is known this is necessary for jointly differentially private algorithms to get non-trivial approximation (e.g., [22]).

To make our approach work in the multi-bidder setting, we need two more ingredients. First, we need to refine our model to control what information are revealed to the bidders in the single-round auctions. At the end of each round, each bidder can observe his own outcome, i.e., whether he wins a copy of the good and his payment. At any round, a bidder's (randomized) bid can depend on his own bids and outcomes in previous rounds, but not those of the other bidders. The information structure plays a crucial role in the argument of bidders' incentives.

Then, it boils down to bound the influence of a bidder's bid on others bidders' outcomes in the same round. This is exactly the main feature of joint differentially privacy. After choosing the reserve price in each round, we run an approximate Vickrey with reserve as follows. First, run the jointly private algorithm of [20] to get a set of roughly $m$ candidate bidders and an approximate Vickrey price. Then, for each candidate bidder, offer a take-it-or-leave-it price that equals the maximum of chosen reserve price and the approximate Vickrey price. The joint privacy of single-round auctions together with the previous argument on the learning process bound how much a bidder's current bid can affect his future utility. Finally, the approximation guarantees of the joint private algorithm ensure that the revenue loss is bounded compared with running Vickrey with the same reserve.

## 1.2 Related Work

There is a vast literature on revenue optimal auction design. We discuss only the most related single-parameter setting. Myerson [33] showed that optimal auctions are (ironed) virtual surplus maximizers. If the bidders' value distributions are i.i.d. and regular, the optimum auction is a Vickrey auction with a reserve price that equals the monopoly price of the distribution. Further, even if distributions are not i.i.d., a Vickrey auction with a suitable reserve still gets a constant approximation [19].

Cole and Roughgarden [10] studied the sample complexity of optimal auctions, and showed upper and lower bounds polynomial in (the inverse of) the error term $\alpha$ and the number of bidders $n$. Bubeck et al. [7] revisited the problem in an online-learning model and introduced algorithms that simultaneously achieve near optimal regret against arbitrary bidder values, improving previous results Blum et al. [6], Blum and Hartline [5], Kleinberg and Leighton [27], and near optimal sample complexity if values are drawn from a underlying distribution. These works implicitly assume myopic bidders so either previous bids are truthful when previous auctions are truthful, or an approximation of the prior distribution can be estimated from the bids in non-truthful previous auctions [34]. This paper takes a more proactive approach of investigating how to design the learning process with the auction to extract meaningful information even if bidders are non-myopic.

Our results build on two lines in differential privacy, namely, differentially private online learning algorithms, and jointly differentially private algorithms. Agarwal and Singh [2] introduced an $(\varepsilon, \delta)$-differentially private algorithm with regret $\tilde{O}(\sqrt{T} + \sqrt{K}/\varepsilon)$ for full information setting (a.k.a. the expert problem), and regret $\tilde{O}(\sqrt{TK}/\varepsilon)$ for bandit setting. Here, $T$ is the number of rounds and $K$ is the number of experts/arms. Independently, there is a same result for bandit setting in [37]. Joint

differential privacy [26] is a a relaxation of differential privacy [15, 14], it can be applied to many combinatorial problems for which no differentially private algorithm gets non-trivial approximation. Hsu et al. [20] introduced the billboard lemma as the cornerstone of joint differential privacy.

**Previous Models of Repeated Auctions with Non-myopic Bidders.** Previous works [11, 24] studied the equilibria of repeated sales when seller cannot commit to a pricing strategy and, thus, must play according to a perfect Bayesian equilibrium. In contrast, we adopt the standard assumption that the seller as a mechanism designer can commit to a strategy upfront. Further, their models assume the same bidder comes every day with value drawn from a prior distribution, while our model assumes no prior, allows different bidders on different days, and the same bidder to have distinct values on different days. Amin et al. [4, 3] considered a stochastic version with the same bidder coming every day, and proposed algorithms with sub-linear regrets, there are better bounds in special case when bidder has the same value on different days [30]. We stress that our model is more general as it assumes no prior and allows different bidders on different days, thus, brings a lot of challenges. Mirrokni et al. [29] also considered repeated auctions with a different model and a different objective compared with ours. They assume prior distributions while we do not, they focus on designing incentive compatible mechanisms while we have a fundamentally different philosophy of achieving non-trivial learning in the equilibrium, which, to our knowledge, has not been considered in the dynamic mechanism design literature before.

**Previous Applications of Differential Privacy in Mechanism Design.** Although differential privacy has been applied to mechanism design before, its role in previous work (e.g, [28, 17, 21, 20, 22]) is fundamentally different from that in ours. First, most previous work achieved approximate incentive compatibility so that misreporting cannot increases a bidder's utility by more than an $\epsilon$ amount. Further, such mechanisms can be coupled with a strictly truthful mechanism to achieve exact incentive compatibility in some specific problems [35]. In contrast, our work uses techniques from differential privacy (rather than the concept itself) to control the influence of a bid in any single round on future utility. Then, we can bound the deviation of a bidder from true value in equilibria (instead of the amount of incentive to deviate). Hence, our approach is conceptually different.

Second, previous work generally used differential privacy to design one-shot mechanisms, while our work considers repeated auctions. Characterizing bidder's behaviors is notoriously hard, a single bidder's deviation in a single round may have the cascading effect of changing the bids of all bidders in subsequent rounds. To this end, we propose to use joint differential privacy as a mean to control information dissemination and, consequently, bidders' behaviors in future rounds in repeated auctions. This is a novel application of joint differential privacy to our knowledge.

In concurrent and independent work, Epasto et al. [17] considered incentive-aware learning and used differential privacy to control the amount of a bidder's deviation from his true value using an approach similar to ours. However, their work focused on a one-shot interaction environment while ours consider repeated auctions. The results are therefore incomparable.

## 2 Preliminary

### 2.1 Single-bidder Model

Let there be a seller who has a fresh copy of the good for sale every day for a total of $T$ days. Exactly one bidder comes on each day, but the same bidder may show up in multiple days. We assume that a bidder can come on at most $\tau$ days for some $\tau \leq T$. Consider the following interactions between the seller and bidders. On each day $t \in [T]$:

1. Seller sets a price $p_t \in [0, 1]$ as a (randomized) function of previous bids $b_1, \ldots, b_{t-1}$.
2. A bidder arrives with value $v_t \in [0, 1]$ and submits a bid $b_t \in [0, 1]$ as a (randomized) function of his value $v_t$, his bids and auction outcomes in the previous rounds that he participates in.
3. Seller observes the bid $b_t$ but not the value $v_t$.
4. Bidder receives the good and pays $p_t$ if $b_t \geq p_t$; nothing happens otherwise.

Here, it is crucial to assume that a bidder does not observe the bids and auction outcomes of the rounds in which he does not participate.

**Rational Bidders.** A bidder's utility in a single round is quasi-linear, namely, $v_t - p_t$ if he gets the good and 0 otherwise. For some discount factor $\gamma \in [0, 1]$, a bidder discounts future utility by at least $\gamma$ and seeks to maximize the sum of discounted utilities. For example, suppose a bidder comes on days $t_1$, $t_2$, and $t_3$. When the bidder considers his strategy on day $t_1$, he would sum up his utilities from all three days, discounting future utility on day $t_2$ by at least $\gamma$, and that on day $t_3$ by at least $\gamma^2$. If $\gamma = 0$, it becomes the model with myopic bidders. If $\gamma = 1$, bidders simply seek to maximize the sum of their utilities. Note that we do not assume that the values of the same bidder must be the same on different days (although they could be).

**Seller.** As a mechanism designer, the seller can commit to a mechanism, i.e., fixing the (randomized) pricing functions $p_1, \ldots, p_T$ upfront. Hence, we shall interpret the $T$-round interactions as a game among the bidders, with the seller designing (part of) the rules. The seller aims to maximize revenue, i.e., the sum of the prices payed by the bidders over all $T$ rounds, denoted as ALG.

We adopt the standard regret analysis of online learning and compare ALG with the optimal fixed price in hindsight, namely, $\max_{p \in [0,1]} p \cdot \sum_{t \in [T]} \mathbf{1}_{v_t \geq p}$.[3] We denote by $\text{OPT}(\{v_t\}_t)$ the revenue of the best fixed price w.r.t. a given sequence of values $\{v_t\}_t$. The *regret* of the algorithm is therefore $\text{OPT}(\{v_t\}_t) - \text{ALG}$. We will further split the regret into two parts as follows in our analysis:

$$\text{OPT}(\{v_t\}_t) - \text{ALG} = \underbrace{\text{OPT}(\{v_t\}_t) - \text{OPT}(\{b_t\}_t)}_{\text{game-theoretic regret}} + \underbrace{\text{OPT}(\{b_t\}_t) - \text{ALG}}_{\text{learning regret}}$$

**Assumptions.** We consider instances that satisfy one of the following two assumptions. The hardness result by Amin et al. [3] implies that no non-trivial regret is possible if neither holds.

- **Large-market:** No bidder participates in a significant portion of rounds, i.e., $\tau = o(T)$.
- **Impatient bidder:** $\gamma$ is (mildly) bounded away from 1, i.e., $\frac{1}{1-\gamma} = o(T)$.

## 2.2 Multi-bidder Model

The model extends straightforwardly to multi-bidder setting. We sketch the model below and highlight a few key assumptions. The seller has $m$ fresh copies of the good for sale every day for a total of $T$ days. $n$ buyers come on each day and a bidder can show up on at most $\tau \leq T$ days. The large-market and impatient bidder assumptions are also applied. Again, a bidder cannot observe the auction outcomes of the rounds in which he does not participate. Further, a bidder cannot observe the auction outcomes of the other bidders, i.e., who gets a copy of the good and how much they pay, even if he participates in that round. Both assumptions on the information structure are crucial for our incentive argument.

Bidders are rational and seek to maximize the sum of their utilities. Seller aims to maximize the total revenue of all $T$ rounds, denoted as ALG. The benchmark, however, is not the revenue of the best fixed arbitrary auction. Instead, we will compare with the revenue of the best fixed auction within a certain family. In this paper, the family of Vickrey auctions with an anonymous reserve price, denote as $\text{OPT}(\{\boldsymbol{v}_t\}_t)$.

In online learning, the algorithm usually has the same strategy space as the offline benchmark. Our model, however, allows the seller to use auctions outside the family of benchmark auctions. We stress that our algorithm uses this flexibility only to implement approximate versions of Vickrey auctions with reserves. Hence, the benchmark is still meaningful.

One can ask the same learning question about other families of auctions, e.g., learning the best anonymous Myerson-type auctions as in Roughgarden and Schrijvers [36] or the best Myerson-type auctions as in Devanur et al. [12]. Extending the techniques in this paper to handle these more complicated auction formats against non-myopic bidders is another interesting future direction.

## 2.3 Differential Privacy Preliminaries

Our techniques rely on the notion of differential privacy by Dwork et al. [15, 14], and its relaxation called joint differential privacy by Kearns et al. [26]. We include the formal definitions as follows.

**Definition 2.1** (Differential Privacy [15, 14]). *An algorithm $A : \mathcal{C}^n \mapsto \mathcal{R}$ is $(\epsilon, \delta)$-differentially private if for all $S \subseteq \mathcal{R}$ and for all neighboring datasets $D, D' \in \mathcal{C}^n$ that differ in one entry, there is $\Pr[A(D) \in S] \le e^\epsilon \Pr[A(D') \in S] + \delta$.*

**Definition 2.2** (Joint Differential Privacy [26]). *An algorithm $A : \mathcal{C}^n \to \mathcal{R}^n$ is $(\varepsilon, \delta)$-jointly differentially private if for any $i$, any $D, D' \in \mathcal{C}^n$ differ only in the $i$-th entry, and any $S \in \mathcal{R}^{n-1}$, there is $\Pr[A(D)_{-i} \in S] \le e^\varepsilon \Pr[A(D')_{-i} \in S] + \delta$.*

## 3 Single-bidder Case: An Overview

Following the treatment of Bubeck et al. [7], we restrict our attentions to prices that are multiples of $\alpha$ and treat each of such prices as an expert in an online learning problem. Let $K = \frac{1}{\alpha} + 1$ denote the number of decretized prices. Consider an expert problem with $K$ experts with the $i$-th expert corresponding to price $(i - 1)\alpha$. We will assume without loss that bids fall into the discretized price set. The bid on day $t$, $b_t$, induces a gain vector $\boldsymbol{g}_t$ such that the gain of the $i$-th expert is $(i - 1)\alpha$ if $b_t \ge (i - 1)\alpha$ and 0 otherwise. That is, $\boldsymbol{g}_t = \left(0, \alpha, 2\alpha, \ldots, \lfloor \frac{b_t}{\alpha} \rfloor \alpha, 0, \ldots, 0\right)$. Further denote the accumulative gain vector up to time $t$ as $\boldsymbol{G}_t = \sum_{j \in [t]} \boldsymbol{g}_j$.

**Theorem 3.1.** *For any $\alpha \in (0, 1)$, there is an online algorithm with regret $\mathcal{O}(\alpha T)$ when $T \ge \tilde{\mathcal{O}}\left(\tau \alpha^{-4}\right)$ under the large market assumption, or $T \ge \tilde{\mathcal{O}}\left(\frac{\alpha^{-4}}{1-\gamma}\right)$ under the impatient bidder assumption.*

Here, we first present a simplified algorithm that gets the regret bound under the stronger assumptions of $T \ge \tilde{\mathcal{O}}\left(\tau \alpha^{-4.5}\right)$ and $T \ge \tilde{\mathcal{O}}\left(\frac{\alpha^{-4.5}}{1-\gamma}\right)$ for intuition. We will then prove the better bounds with a more complexed algorithm (similar key ideas) in full version.

### 3.1 (Simplified) Algorithm

**Tree-aggregation.** The simplified algorithm is a privacy-preserving version of the followed-the-perturbed-leader algorithm based on the tree-aggregation technique [16, 9]. Since our analysis needs to make use of the structure of the algorithm, it is worthwhile to devote a few paragraphs to formally define the tree-aggregation subroutine.

Suppose we have $T$ elements (the experts' gains) and need to calculate the cumulative sum of elements from 1 to $t$ for any $t \in [T]$ in a differentially private manner. The naïve approach simply calculates the cumulative sums and add, say, Gaussian noise, to each of them. Since an element may appear in all $T$ cumulative sums, the noise scale is $\tilde{\mathcal{O}}(\sqrt{T}/\epsilon)$ by a standard argument. Instead, the tree-aggregation technique calculates $T$ partial sums such that (1) each element appears in at most $\log T$ partial sums, and (2) each cumulative sum is the sum of at most $\log T$ partial sums. This technique significantly reduces the noise scale to $\tilde{\mathcal{O}}(1/\epsilon)$.

Next, we explain how to design the partial sums. Consider any $t \in [T]$ with binary representation $(t_{\log T} \ldots t_1 t_0)_2$, i.e., $t = \sum_{j=0}^{\log T} t_j \cdot 2^j$. Let $j_t$ be the lowest non-zero bit. The $t$-th sum is over

$$\Lambda_t = \left\{t - 2^{j_t} + 1, t - 2^{j_t} + 2, \ldots, t - 1, t\right\}.$$

To compute the sum of the first $t$ elements, it suffices to sum up the following sets of partial sum obtained by removing the non-zero bits of $t$ one by one from lowest to highest:

$$\Gamma_t = \left\{t' \ne 0 : t' = t - \sum_{j=0}^{h-1} t_j 2^j, h = 0, 1, \ldots, \log T\right\}$$

Then, we have $[t] = \cup_{j \in \Gamma_t} \Lambda_j$.

For example, suppose $t = 14 = 1 \cdot 2^1 + 1 \cdot 2^2 + 1 \cdot 2^3$. Then, we have $\Lambda_t = \{13, 14\}$ and $\Gamma_t = \{14, 12, 8\}$. The tree-aggregation subroutine is given in Algorithm 1.

The usual description of tree-aggregation (e.g., [2]) computes the partial sum $\boldsymbol{A}_t$ in one-shot at step $t$ while ours considers $\boldsymbol{A}_t$'s as internal states that are maintained throughout the algorithm. Both descriptions result in the same algorithm but ours is more convenient for our proof.

**Lemma 3.2** (Jain et al. [25]). *The final values of the internal states $\boldsymbol{A}_t$'s are $(\epsilon, \delta = \frac{\epsilon}{T})$-differentially private with $\sigma = \frac{8\sqrt{K}}{\varepsilon} \log T \sqrt{\ln \frac{\log T}{\delta}}$.*

---

**Algorithm 1** Tree-aggregation

---
1: **input:** dimension $K$, gain vector $\mathbf{g}_t \in [0,1]^K$ of each round $t$, noise scale $\sigma$
2: **internal states:** noisy partial sum $\boldsymbol{A}_t$ for $t \in [T]$
3: **initialize:** $\boldsymbol{A}_t = \boldsymbol{\mu}_t$ for all $t \in T$, with $\mu_{tj}$'s i.i.d. from normal distribution $\mathcal{N}(0, \sigma^2)$.
4: **for** t = 1, 2, ..., T **do**
5:     Receive $\boldsymbol{g}_t$ as input.
6:     Let $\boldsymbol{A}_j = \boldsymbol{A}_j + \boldsymbol{g}_t$ for all $j$ s.t. $t \in \Lambda(j)$.
7:     **Output** $\tilde{\boldsymbol{G}}_t = \sum_{j \in \Gamma_t} \boldsymbol{A}_j + \boldsymbol{\nu}_t$, with $\nu_{tj}$'s i.i.d. from $\mathcal{N}\big(0, (\log T + 1 - |\Gamma_t|)\sigma^2\big)$.
8: **end for**

---

We need a slightly stronger version that is a simple corollary.

**Lemma 3.3.** *Fix any* $t_0 \in [T]$, *the values of the internal states* $\boldsymbol{A}_t$'s *at the end of round* $t_0$ *are* $(\epsilon, \delta = \frac{\epsilon}{T})$*-differentially private for bids on or before day* $t_0$ *with* $\sigma = \frac{8\sqrt{K}}{\varepsilon} \log T \sqrt{\ln \frac{\log T}{\delta}}$.

*Proof.* The values of $\boldsymbol{A}_t$'s at the end of round $t_0$ are effectively the values if subsequent gain vectors are all zero. Hence, the lemma follows as a corollary of Lemma 3.2. $\square$

**Online Pricing Algorithm.** Our algorithm (Algorithm 2) is a variant of the privacy-preserving online learning algorithm of [2]. It uses tree-aggregation as a subroutine for maintaining an noisy version of the cumulative gains of each price. On each day $t$, with some small probability it picks the price randomly; otherwise, it picks the price with the largest noisy cumulative gain in previous days, i.e, the largest entry of $\tilde{\boldsymbol{G}}_{t-1}$. In addition, due to step 7 in Algorithm 1, we can obtain that $\tilde{G}_{tj} - G_{tj}$ follows $\mathcal{N}(0, (\log T + 1)\sigma^2)$ for any $t \in [T]$ and any $j \in [K]$.

---

**Algorithm 2** Online Pricing (Single-bidder Case)

---
1: **parameters:** regret parameter $\alpha$, $K = \frac{1}{\alpha} + 1$, privacy parameter $\epsilon$, $\delta = \frac{\epsilon}{T}$.
2: **initialize** tree-aggregation (Alg. 1) with $\sigma = \frac{8\sqrt{K}}{\varepsilon} \log T \sqrt{\ln \frac{\log T}{\delta}}$.
3: **for** $t = 1, \ldots, T$ **do**
4:     With probability $\alpha$, pick $j \in [K]$ uniformly at random.
5:     Otherwise, pick $j$ that maximizes $\tilde{\boldsymbol{G}}_{(t-1)j}$.
6:     Set price $(j-1)\alpha$.
7:     Observe bid $b_t$ and, thus, the gain vector $\boldsymbol{g}_t$; update tree-aggregation.
8: **end for**

---

## 3.2 Bound Learning Regret

Lemma follows from Theorem 8 of [1].

**Lemma 3.4.** *Consider running Algorithm 2 without step 4. Then, the learning regret w.r.t. the best fixed discretized price is at most* $\mathcal{O}\big(\sqrt{\log K}(\sigma\sqrt{\log T} + \frac{T}{\sigma\sqrt{\log T}})\big)$.

**Corollary 3.5.** *The learning regret of Algorithm 2 is* $\mathcal{O}\big(\sqrt{\log K}(\sigma\sqrt{\log T} + \frac{T}{\sigma\sqrt{\log T}}) + \alpha T\big)$.

*Proof.* Running Algorithm 2 with step 4 increases the regret by at most $\alpha T$. Further note that the regret w.r.t. to the best fixed discretized price differs from the actual regret by at most $\alpha T$. $\square$

## 3.3 Bounding Game-theoretic Regret: Stability of Future Utility

We prove the bound by showing each bidder won't deviate far, i.e, $|b_t - v_t| \leq 2\alpha$ as Lemma 3.7 (proof in full version). This is because lying is lower bounded in the current round by step 4, and the extra utility in the future is upper bounded by Lemma 3.6.

**Lemma 3.6** (Stability of Future Utility)**.** *For any bidder and any day* $t$ *on which he comes, the bidder's equilibria utilities in subsequent rounds in the subgames induced by different bids on day* $t$ *differ by at most an* $e^\epsilon$ *multiplicative factor plus a* $\delta T$ *additive factor.*

Readers may think of perfect Bayesian equilibrium as a concrete solution concept to understand the lemma. However, note that the problem is defined without Bayesian priors, and the lemma holds for a much more general, yet non-standard, solution concept. Our proof shows that no matter what a bidder's belief is on the other bidders' values and strategies, assuming she always plays best-response under her belief in subsequent rounds, the future utility differ by at most an $e^\epsilon$ multiplicative factor.

This is the main argument of our approach. First consider a seemingly intuitive yet incorrect proof. Since tree-aggregation is differentially private, the online pricing algorithm is also private treating the bid on each day as an entry of the dataset. The lemma holds because changing the bid on a day leads to a neighboring dataset and, thus, the probability of any subset of future outcomes does not change much. This is incorrect because subsequent bids are controlled by strategic bidders. Changing the bid on a day does not result in a neighboring dataset in general.[4]

*Proof.* We shall abuse notation and refer to the bidder that comes on day $t$ as bidder $t$. Fix any bidder $t$'s strategy for subsequent days (after round $t$). That is, fixed the (randomized) bidding function on any subsequent day $t'$ as a function only on his bids and auction outcomes between day $t$ (exclusive) and $t'$ (exclusive). Let us consider the resulting utilities for bidder $t$ in the subgames induced by two distinct bids on day $t$. Note that the other bidders' subsequent strategies will be the same in the subgames since they cannot observe what happens on day $t$. We shall interpret the execution of the online pricing algorithm, i.e., the algorithm together with the bidders' strategies in subsequent rounds, after round $t$ as a post-processing on the internal states of the tree-aggregation algorithm at the end of day $t$ and, thus, is $(\epsilon, \delta)$-differentially private due to Lemma 3.3. Therefore, the utilities of any fixed subsequent strategy of bidder $t$ in the two subgames differ by at most an $e^\epsilon$ multiplicative factor plus a $\delta T$ additive factor. The lemma then follows by the equilibria condition that bidder $t$ employs the best subsequent strategy in any subgame. In fact, if the bidder on day $t$ uses two different *arbitrary* strategies in subsequent subgames depending on his different bids on day t, indeed it is impossible to bound the change in his utility. However, we assume that the bidder is rational so his strategy in subsequent rounds satisfies equilibria conditions. Then, it suffices to show that for any fixed subsequent strategy, the future utility does not change much, because the equilibria utility is simply taking a max over all possible subsequent strategies. $\square$

**Lemma 3.7.** $\forall t$, we have $|b_t - v_t| \leq 2\alpha$, and then the game-theoretic regret is bounded by $2\alpha T$ for

- $\alpha = (4\tau\epsilon)^{1/3}$ under the assumption of large market; or
- $\alpha = (\frac{4\epsilon}{1-\gamma})^{1/3}$ under the assumption of impatient bidders.

### 3.4 Bounding Total Regret

We prove the regret under the large market assumption. The case of impatient bidders is almost identical. Putting together Corollary 3.5 and Lemma 3.7, the regret of Algorithm 2 is at most $\mathcal{O}\big(\sqrt{\log K}(\sigma\sqrt{\log T} + \frac{T}{\sigma\sqrt{\log T}}) + \alpha T\big)$ for $\alpha = (4\tau\epsilon)^{1/3}$. This means that we shall set $\epsilon = \Theta(\alpha^3/\tau)$ and, thus, $\sigma = \tilde{\Theta}\big(\sqrt{K}/\varepsilon\big) = \tilde{\Theta}\big(\tau\alpha^{-3.5}\big)$. So the 2nd term in the above regret bound is negligible compared to $\alpha T$. The regret bound becomes $\tilde{\mathcal{O}}\big(\tau\alpha^{-3.5}\big) + \mathcal{O}\big(\alpha T\big) \leq \mathcal{O}\big(\alpha T\big)$ if $T \geq \tau\alpha^{-4.5}$.

## 4 Multi-bidder Case: An Overview

We take the same online learning formulation as in the single-bidder case, treating each discretized price that is a multiple of $\alpha$ between 0 and 1 as an expert. Expert $j$'s gain on any day is the revenue of Vickrey auction with reserve price $(j-1)\alpha$ w.r.t. the bids on that day. We sketch the main ideas below, and present the proof in full version.

**Theorem 4.1.** *For any $\alpha \in (0, 1)$, our algorithm runs an approximate version of Vickrey with an anonymous reserve price on each day with regret $\leq \alpha m T$ against the best fixed reserve price if:*

1. *$T \geq \tilde{O}\big(\frac{\tau n}{m\alpha^{4.5}}\big)$, and $m \geq \tilde{O}(\frac{\sqrt{\tau n}}{\alpha^3})$ given large market; or*
2. *$T \geq \tilde{O}\big(\frac{n}{(1-\gamma)m\alpha^{4.5}}\big)$, and $m \geq \tilde{O}(\frac{\sqrt{n}}{\sqrt{1-\gamma}\alpha^3})$ given impatient bidders.*

**Algorithm.** With some small probability we randomly pick a subset of bidders and offer each of them a copy of the good with a random price to ensure lying is costly in the current round. We pick the reserve price on each day using follow-the-perturbed-leader implemented with tree-aggregation. Simply running Vickrey with the chosen reserve price does not guarantee stability of future utility, however, because a bidder's current bid can now affect other bidders' subsequent bids through the allocations and payments in the current round. Instead, we use a private allocation algorithm of [20] to get a set $S$ and a price $p$ that are approximations of the set of top-$m$ bidders and Vickrey price.

---

**Algorithm 3** Online Pricing (Multi-bidder Case)

---

1: **input:** regret parameter $\alpha$, $K = \frac{1}{\alpha} + 1$, privacy parameter $\epsilon$, $\delta = \frac{\epsilon}{T}$, $E = \tilde{\mathcal{O}}\left(\frac{1}{\alpha^2 \epsilon}\right)$.
2: **initialize** tree-aggregation with noise scale $\sigma = \frac{8\sqrt{K}\log T}{\varepsilon}\sqrt{\ln \frac{\log T}{\delta}}$.
3: **for** $t = 1, \ldots, T$ **do**
4:     With probability $\alpha$, pick a subset $S \subseteq [n]$ of size $m$ and $j \in [K]$ uniformly at random.
5:     Otherwise:
6:         Pick $j_1$ that maximizes $\tilde{\boldsymbol{G}}_{(t-1)j}$ from tree-aggregation.
7:         Run PMatch($\alpha, \rho = \alpha, \epsilon$) [20] to get a set $S$ of $\leq m - E$ bidders and a price $p = j_2 \alpha$.
8:         Let $j = \max\{j_1 - 1, j_2\}$.
9:     Offer a copy of good to each $i \in S$ at price $(j - 1)\alpha$.
10:    Observe bid vector $\boldsymbol{b}_t$; update tree-aggregation with the normalized gain vector $\frac{1}{m}\boldsymbol{g}_t$.
11: **end for**

---

**Stability of Future Utility.** It is similar to the single-bidder case. Consider the bidder on some fixed day $t$ and the subgames induced by two distinct bids on day $t$. Fix any subsequent strategy of bidder $t$. The subsequent execution of the algorithm can be viewed as a post-processing on the internal states of tree-aggregation together with the other bidders' memberships w.r.t. $S$ and the price $p$ on day $t$, which are differentially private due to Lemma 3.3 and the privacy property of PMatch in [20].

**Regret.** The main extra ingredient from single-bidder is the revenue of approximate implementation of Vickrey with reserve does not deviate much from that of the exact implementation.

**Lemma 4.2** (Hsu et al. [20]). *The set of bidders $S$ and the price $p$ satisfy:*

1. *$m - 2E \leq |S| \leq m - E$;*
2. *all bidders in $S$ have values at least $p - \alpha$;*
3. *at most $E$ bidders outside $S$ have values at least $p$.*

**Lemma 4.3.** *For any $j^* \in [K]$, the revenue of running Vickrey with reserve $p^* = (j^* - 1)\alpha$ is no more than that of running steps 7-9 in Algorithm 3 with $j_1 = j^*$ plus $\mathcal{O}(E + \alpha m)$.*

*Proof.* Suppose $p' = j'\alpha$ is the $(m+1)$-th highest bid. The winners in Vickrey pays $\max\{p^*, p'\}$. With $j_1 = j^*$, $(j_1 - 2)\alpha = p^* - \alpha$. Claims 1 and 3 of Lemma 4.2 imply $p \geq p'$ and, thus, $(j_2 - 1)\alpha \geq p' - \alpha$. Hence, the price offered in step 9 is at least $\max\{p^*, p'\} - \alpha$. It remains to show the number of sales by steps 7-9 is less than that of Vickrey by at most $\mathcal{O}(E)$. If $(j_1 - 2)\alpha$ is offered in step 9, then the number of sales by the algorithm is at least that of Vickrey with reserver $p^*$ minus $E$ due to claim 3 of Lemma 4.2. If $(j_2 - 1)\alpha = p - \alpha$ is offered in step 9, then the number of sales is at least $m - 2E$ due to claim 1 and 2 of Lemma 4.2. Hence, it is less than the number of sales of Vickrey by at most $2E$. In both cases, the lemma follows. $\square$

## Footnotes

[2]Here, $T \geq \tilde{\mathcal{O}}(\alpha^{-4})$ means that for any $\alpha$, there exits some function $h(\alpha) = \tilde{\mathcal{O}}(\alpha^{-4})$ such that the theorem holds when $T \geq h(\alpha)$.

[3]Note that a bidder's best strategy against a fixed price is truthful bidding.

[4]Although other bidders do not see what happens on day $t$, thus, will employ the same strategies in subsequent days, the actual bids are affected also by the seller's subsequent prices, which is affected by the bid on day $t$.

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
