[Reviews · NeurIPS 2018]

Reviewer 1



This paper studies the problem of lerning optimal reserve price in repeated auctions. The paper focus on non-myopic bidders (i.e. bidders who show up in more than 1 round and strategically reason about how the current bid may affect gain in future rounds). The paper shows algorithm that obtain non-trival regret if one of two following conditions are satisfied: 1)large market: each bidder only participate in sublinear number of rounds. 2)impatient bidders: bidders have discount factor o(1). Overall, I like the results of this paper. The myopic assumption of many previous work is not very realistic and avoids the game-theoretic difficulty in the problem. The split of regret into game-theoretic regret and learning regret is a much better way to think about the problem. I also like the connection between dynamic mechanism design and differential privacy/jointly diffential privacy shown in the paper. Comments: 1) The paper uses the big-O notation in the following way: f(x) >= O(g(x)). I feel this is confusing as O(g(x)) is a subset of functions. It could mean there exists h(x) = O(g(x)) such that f(x)>=h(x). This is meaningless as you can always pick h(x) = 0. I guess what the authors actually mean is there exists h(x) = \Theta(g(x)) such that f(x)>=h(x). 2) Page 3: in the statement of Informal Theorem 2, despite the big-O notation problem mentioned above, it's hard to parse what regret is achieved. Maybe the statement should just tell what regret is achieved when picking the best alpha. 3) In the informal statements, it's a bit misleading to not mention the tradeoff between regret and discount factor or number of rounds of a bidder. The statements of informal theorems are only true when the discount factor is constant away from 1 or each bidder participates in constant number of rounds. Reply to Author Response: Some additional comments for comment 3): yeah it would be nice to make discount factor (or number of rounds of a bidder) explicit in the informal theorems. I am thinking you can remove alpha in the informal theorem and directly state the optimal regret. For example, informal theorem 1 for discount factor would have regret roughly T^{3/4} (1-\gamma)^{-1/4}.

Reviewer 2



Summary The paper studies the problem of learning the optimal reserve price to use in a repeated auction environment in order to maximize revenue. Prior work has focused on optimizing revenue in cases where the distribution of buyer values is unknown but the seller has access to samples. However, performing these in settings where bidders are non-myopic introduces incentive issues as bidders may want to misreport and lower the bids to avoid future increases in prices. This work explores settings where bidders are non-myopic and the seller can commit in advance in a process through which the prices will be updated. It proposes a generic method based on techniques from differential privacy that achieves low regret against the optimal price in hindsight for non-myopic bidders. The method works under the assumption that either the market is large (no bidder appears many times) or that bidders discount future utilities. The main results are that: - For a single bidder per round, aT regret is achievable for any a>0 when T > a^(-4). - For n bidder per round with m copies of items, amT regret is achievable for any a>0 when T > n/(m a^4.5) and m > sqrt(n)/a^3. To establish the results, the regret is broken down to learning regret and game theoretic regret. Learning regret is the best revenue achievable by a fixed price given the observed bids minus the revenue of the algorithm, while the game theoretic regret is the revenue difference when pricing with respect to the values vs the bids. The algorithm used to achieve those results, performs at every round a follow-the-perturbed-leader strategy but also uses a uniformly random price with small probability. The follow-the-perturbed-leader strategy guarantees that the learning regret is small, but also that future rounds are not significantly affected by current bids. At the same time, the uniform random price forces bidders to bid close to their true value (similarly to the "gap" concept of [Nissim'12]). This allows bounding the game-theoretic loss. Comments - One issue with the presentation of the results is that the bidder strategies are not formally defined. At many points it is argued that bidders will bid close to their true value in equilibrium. However, the equilibrium notion that is used is not defined anywhere and I find this whole discussion quite vague. - Another weakness of the work is that no lower bounds are provided. It is hard to judge how much revenue is left on the table without knowing how much a seller should expect to make in such a setting. - For the case with many bidders per round, it seems kind of strange that the number of copies m needs to grow when aiming for lower regret. Is there some inherent limitation or intuition for why it is hard to obtain results for a single item (m=1)? - A branch of work that deals specifically with repeated interactions with bidders is the field of dynamic mechanism design e.g. the work of Mirrokni et al. "Non-clairvoyant Dynamic Mechanism Design". It would be nice to compare the setting and results to this literature. Evaluation Overall, I like the proposed algorithm as it is quite simple and is able to capture complicated settings with many bidders participating in many rounds. The paper manages to show sublinear regret bounds but there is still room for improvement to match the sqrt(T) regret lower bound. The connection to joint-differential privacy is quite interesting but seems to have limitations for many bidders. It is unclear whether an alternative approach can yield non-trivial regret bounds for m=1.

Reviewer 3



Summary: This paper studies the interesting and important problem of online learning for pricing problems when the buyers are strategic. There is a seller who interacts with a set of buyers over a series of time steps. The seller has a fresh copy of the good for sale every day. She sets a price at the beginning of each day. Then, a bidder arrives and submits a bid: If the bid is higher than the price of the day, he gets the item and pays the price. The question is: how should the seller set the price? He could use an off-the-shelf no-regret learning algorithm. However, if the same buyer returns several times, she might be incentivized to misreport her value in order to fool the learning algorithm into giving her a lower price. Much of the existing literature on this topic assumes the buyers are myopic, which means they do not take future rounds into account when bidding in the current round. This paper does not make this assumption. This paper presents a no-regret pricing algorithm that satisfies a relaxed version of incentive compatibility. It works under one of two assumptions: Either the bidders discount utility in future rounds by some discount factor bounded away from 1, or any one bidder comes in only a small fraction of the rounds. Their algorithm discretizes the set of prices and treats each discretized point as an expert. At each round, each expert’s reward equals their corresponding price if the bidder bid above their price, and zero otherwise. The algorithm uses an existing differentially privacy tool called “tree-aggregation” to keep track of noisy estimates of each expert’s gain. The algorithm picks a reserve on each day using follow-the-perturbed-leader. The authors generalize to multiple bidders and multiple units of items for sale, too. Comments: According to the authors, one of the key insights is to use joint differential privacy rather than differential privacy itself. However, joint differential privacy is not mentioned in Section 3 as far as I can tell, which is a little confusing. Perhaps this is because Section 3 is a simplified version of their main algorithm. In my experience, take-it-or-leave-it auctions, which are commonly referred to as posted price mechanisms, do not take bids as input. The buyer either buys the good and pays the price, or does not buy the good. In what scenario would he be required to submit a bid? The authors write that online learning for auction design has only been studied “very recently” (line 16), which is not true. For example, see the following paper and subsequent work. Kleinberg and Leighton. The Value of Knowing a Demand Curve: Bounds on Regret for Online Posted-Price Auctions. FOCS 2003. The authors write that “typical mechanism design approach may seek to design online learning mechanisms such that truthful bidding forms an equilibrium (or even better, are dominating strategies) even if bidders are non-myopic, and to get small regret given truthful bids. However, designing truthful mechanisms that learn in repeated auctions seems beyond the scope of existing techniques.” Can the author comment on what they mean by “beyond the scope of existing techniques?” Isn’t the point of research to come up with new techniques? :-) I think it’s fine to pursue the path the authors chose, but I’d recommend finding a better argument. In the related work, the authors write that “most previous work achieved approximate incentive compatibility so that misreporting cannot increases a bidder’s utility by more than an epsilon amount.” In contrast, the authors’ relaxed version of incentive compatibility ensures that “bidders would only submit ‘reasonable’ bids within a small neighborhood of the true values.” What are the pros and cons of both relaxations to incentive compatibility? This might be helpful for understanding why there ought to be another paper on this topic. =====================After rebuttal===================== Thanks for the clarifications!